# Forecasting dryland vegetation condition months in advance through satellite data assimilation

Siyuan Tian [1,2], Albert I.J.M. Van Dijk [2], Paul Tregoning [1] & Luigi J. Renzullo [2]

Dryland ecosystems are characterised by rainfall variability and strong vegetation response to changes in water availability over a range of timescales. Forecasting dryland vegetation condition can be of great value in planning agricultural decisions, drought relief, land management and fire preparedness. At monthly to seasonal time scales, knowledge of water stored in the system contributes more to predictability than knowledge of the climate system state. However, realising forecast skill requires knowledge of the vertical distribution of moisture below the surface and the capacity of the vegetation to access this moisture. Here, we demonstrate that contrasting satellite observations of water presence over different vertical domains can be assimilated into an eco-hydrological model and combined with vegetation observations to infer an apparent vegetation-accessible water storage (hereafter called accessible storage). Provided this variable is considered explicitly, skilful forecasts of vegetation condition are achievable several months in advance for most of the world's drylands.

---

[1] Research School of Earth Sciences, Australian National University, Canberra 2601 ACT, Australia. [2] Fenner School of Environment & Society, Australian National University, Canberra 2601 ACT, Australia. Correspondence and requests for materials should be addressed to S.T. (email: siyuan.tian@anu.edu.au)

The majority of ecosystems globally are persistently or seasonally limited by water availability[1]. Dryland vegetation responds to rainfall variability in contrasting ways, depending on the timescale of rainfall variability and the way that this interacts with soil hydraulic properties and vegetation rooting patterns[2–4]. Together, these factors determine the vegetation-accessible water storage capacity. Variations in water availability affect the growth and condition of grazing land, dryland crops and planted forests, as well as native vegetation. Vegetation condition, in turn, affects fire risk[5] and soil health[6] and can contribute to heatwaves through land–atmosphere feedback processes[7]. Forecasting vegetation condition in response to water availability months ahead would therefore be of great value for timely mitigation of such impacts.

Unfortunately, for most of the world's dryland areas, rainfall is very unpredictable[8] or with low forecast skill at monthly timescale and beyond. Most climate modes do not persist very long and those that do, such as the El Niño Southern Oscillation and Indian Ocean Dipole, tend to achieve comparatively less skill in drier regions[9]. However, water stored at and below the surface provides a source of forecasting skill that can be more influential over longer periods, as has been demonstrated for streamflow[10,11]. Soil moisture has a memory that persists for weeks to months, depending on the relative magnitude of vegetation-accessible storage and precipitation variability[2,6]. This suggests the potential to use root-zone soil water availability to forecast vegetation condition at large scale. So far, this potential remains unexplored. This is likely in part because of the lack of accurate knowledge of accessible storage capacity and the low fidelity of hydrological models in estimating vertical moisture distribution[12–14]. In weather forecasting, assimilation of atmospheric satellite observations mitigates model deficiencies to provide better estimates of system state, and this has been the main driver of remarkable enhancements of weather forecast skill and lead time[15]. Here, we demonstrate that data assimilation can produce similar benefits in ecohydrological forecasting.

Satellite remote sensing has been pivotal to deepening our understanding of water availability and climate change at regional-to-global scale, and has helped to advance predictive models and decision making[16]. However, satellite observations of water presence are limited to either the surface (up to 5 cm for soil moisture, e.g., Soil Moisture and Ocean Salinity (SMOS) mission) or total water column (Gravity Recovery and Climate Experiment (GRACE) mission). The quantification of the vertical distribution of water storage is extremely difficult over large spatial and time domains due to the lack of direct measurement of root-zone soil moisture and groundwater storage. The accuracy of soil moisture or groundwater storage estimates separated from total water storage is limited without ancillary data and the consideration of data uncertainties[17]. We assimilated MODIS (Moderate Resolution Imaging Spectroradiometer) satellite instrument-derived surface water extent[18], SMOS near-surface soil moisture[19] and GRACE total column water storage[20] into a global ecohydrological model[11] and estimated the vertical distribution of water at the surface[18], in the near-surface soil, shallow root zone (<1 m), deep root zone (>1 m) and in groundwater[21] (see Methods). Satellite-derived vegetation greenness (i.e., the Normalised Difference Vegetation Index (NDVI)) was used as a simple but powerful measure of vegetation condition. In areas of low-density vegetation, NDVI is generally a strong proxy of vegetation cover fraction, leaf area and biomass. The average seasonal cycle of greenness is inherently predictable and was subtracted from the observations, resulting in greenness anomalies. The monthly greenness anomalies, on the one hand, and anomalies in water storage integrated over different depths, on the other, were used to develop a simple forecast model. A skilful lead time was defined as the forecast period over which rank correlation ($\rho$) between accessible storage and greenness remained relatively high ($\rho > 0.60$). The results were analysed as a function of climate dryness at each location, defined as the long-term average fraction of months for which potential evapotranspiration exceeds precipitation (see Methods).

We find that larger accessible storage broadly corresponds with slower decay in forecast skill. Vegetation conditions in the majority of global dryland can be forecast 3 months in advance from accurate estimates of current soil water availability.

## Results

**Vegetation response to water stores**. Vegetation in dry climatic zones with dryness value over 0.8 (Fig. 1a) generally shows greater accessible storage (>100 mm) and less reliance on surface water than vegetation in more humid zones (Fig. 1b). For example, vegetation in up to 70% of the more humid areas (dryness index 0.4–0.6) shows greater response to the shallow soil water with less than 50 mm of accessible storage, while more than 65% of dryland vegetation (dryness 0.7–1.0) appears to have access to water at >1 m below the surface. With increasing dryness, accessible storage is an increasingly strong predictor of future vegetation greenness (Fig. 1c). Naturally, forecast skill decayed over time, but skilful forecasts were often still achieved as long as 3 months ahead. In such areas, 80% of the vegetation appeared to have access to deeper soil moisture. Thus, prediction lead time can be broadly interpreted as a measure of vegetation access to deep water stores.

Alternative forecasts were also developed using an antecedent precipitation index and remotely sensed near-surface soil moisture or total water storage, but these typically provided skilful vegetation forecasts for no more than 1 or 2 months (Fig. 1d). Skilful forecasts using soil water availability from satellite observations or model simulations could be achieved for no more than 20% of the vegetated arid area (dryness >0.6). Estimates of accessible storage derived through assimilation of satellite observations led to considerably better forecasts; skilful forecasts were provided for a greater fraction of area for all dryness categories. This is the result of the integration of satellite observations of water present near the surface and at greater depth with the process understanding encoded in the ecohydrological model.

Particularly skilful forecasts and long lead times of over 5 months were found for interior Northern Australia, corresponding with dry but dominantly perennial grassland and shrubland showing relatively high accessible storage (c. 200 mm) (Fig. 2). Positive spatial correlation between accessible storage and lead time is also evident in other regions. Vegetation condition forecasts in sub-humid and humid regions (dryness <0.5) are generally less robust, particularly towards higher latitudes. This is as would be expected given that temperature and radiation will be equal or stronger drivers of greenness than water availability[1,22]. Some part of the forecast skill can be attributed to the correlation between the average seasonal cycles of water storage and greenness, particularly in monsoon climates. This source of forecast skill can be exploited in the absence of water storage information (see Methods) and can be subtracted from overall skill to highlight regions where water storage information provides an important contribution to forecast skill (Fig. 3a). The best performing between the climatology forecast and persistence forecast at each pixel was selected and compared with our result. Significant improvements were found over regions vulnerable to droughts and poorly predictable with seasonal patterns.

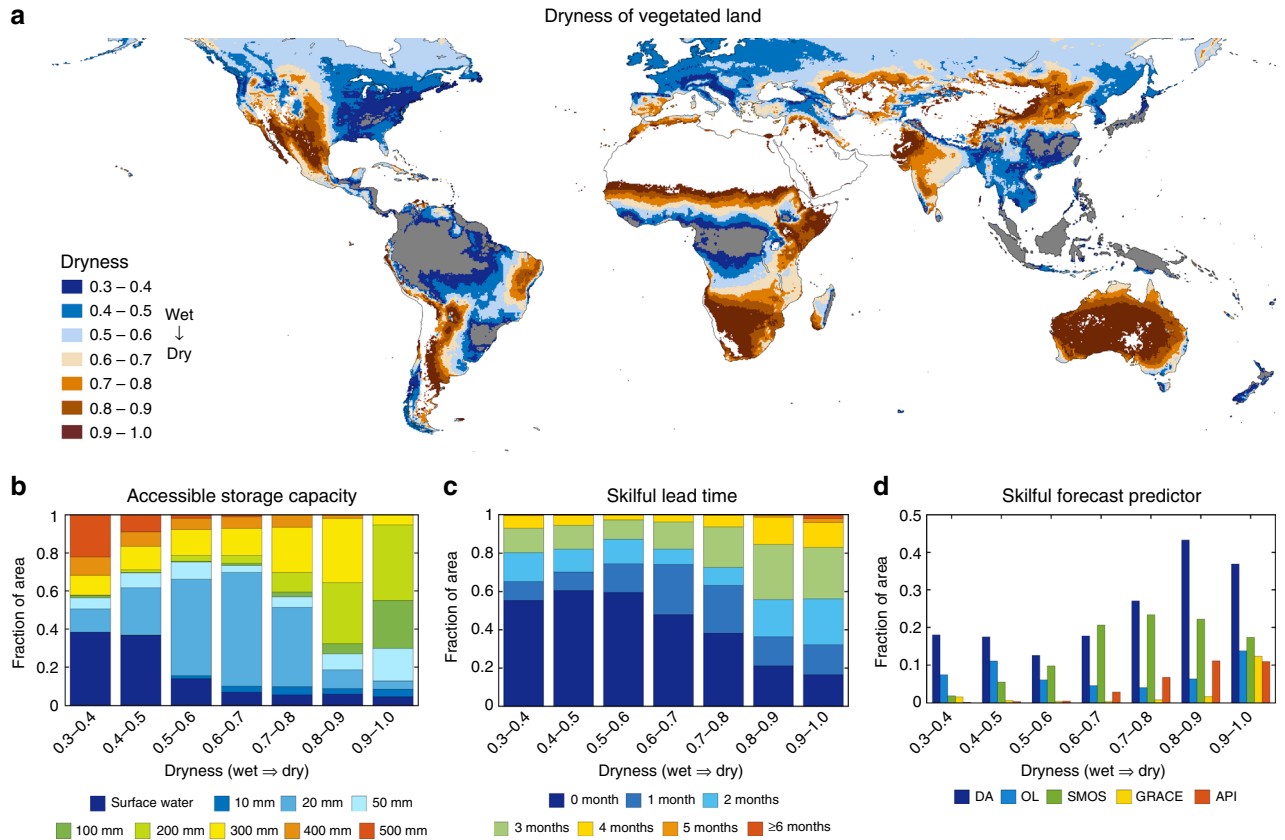

**Fig. 1** Accessible storage and vegetation dynamics prediction skill. Relationship between water availability over different integration depths and vegetation greenness anomalies over humid to arid regions with dryness indices from 0.3 to 1.0. **a** Distribution of global drylands; areas with minimal vegetation (maximum Normalised Difference Vegetation Index (NDVI) <0.25) and generally high water availability were masked out in white and grey, respectively. **b** Fraction of area for accessible storage capacity in mm (surface water or below-surface) at different dryness levels. **c** Fraction of area for the number of months for which skilful ($\rho > 0.6$) forecasts were achieved in different dryness levels. **d** Fraction of area for which skilful forecasts were possible 3 months in advance using data assimilation (DA), compared to those achieved using only open-loop model results without any assimilation of satellite observations (OL), using satellite-derived near-surface soil moisture (Soil Moisture and Ocean Salinity (SMOS)), using total water storage (Gravity Recovery and Climate Experiment (GRACE)) and using an index calculated from antecedent precipitation only (Antecedent Precipitation Index (API))

**Forecasts of dryland vegetation condition**. Case studies for southern California, central Queensland and the Horn of Africa illustrate features of the forecasts. Skilful 1-month and 3-month forecasts of vegetation response to drought conditions from 2011 until 2014 in California were made (Fig. 3b, c). The developing impacts of a multi-year drought from 2012 onwards in Queensland, Australia, were also forecast 1 month and 3 months ahead (Fig. 3d, e). Superior skill to forecast the severe drought in the Horn of Africa from 2011 to 2012 was demonstrated and cannot be achieved with the traditional monitoring forecasts even 1 month ahead (Fig. 3f, g). Significant improvements with an increase in correlation of more than 0.2 were achieved with longer lead time compared with NDVI-climatology forecasts. Forecasts using accessible storage showed a slower decay of forecast skill than NDVI-based forecasts by more than 0.1 units and maintained a correlation of ~0.8 in 3-month forecasts. A further increase in the historical assimilation period should help to further improve forecast model skill (see Methods).

## Discussion

The interplay between soil water availability and the intensification of drought differs with soil depth and aridity[23]. Our study used plant-accessible storage across dryland areas to explore the relationship between water availability and dryland vegetation condition. The accessible storage capacity inferred here is empirically defined and may be less than the total moisture storage that can be accessed by the deepest-rooted individuals within the ecosystem. Rather, our results indicate the soil water store that empirically best predicts vegetation anomalies for the visually dominant ecosystem component as observed by remote sensing. Nonetheless, in semi-arid to arid regions we found spatial patterns that are very similar to previously reported root-zone storage capacity and rooting depths[14,24,25].

Our estimates of the accessible storage combine soil water dynamics information captured by multiple satellite sensors through data assimilation. A stronger response of vegetation greenness to water availability was found using accessible storage, when compared against water availability derived from only satellite observations or the ecohydrological model, and results from previous studies[26–29]. Our findings suggest that incorporating current soil water availability, can significantly improve the accuracy of vegetation condition forecasts 3 months in advance for the majority of drylands globally. Such forecasts can help to improve drought early warning system and reduce economic and environmental impacts. This capacity may become even more important in the context of projected increases in the occurrence and severity of drought under climate change in some regions[30–32].

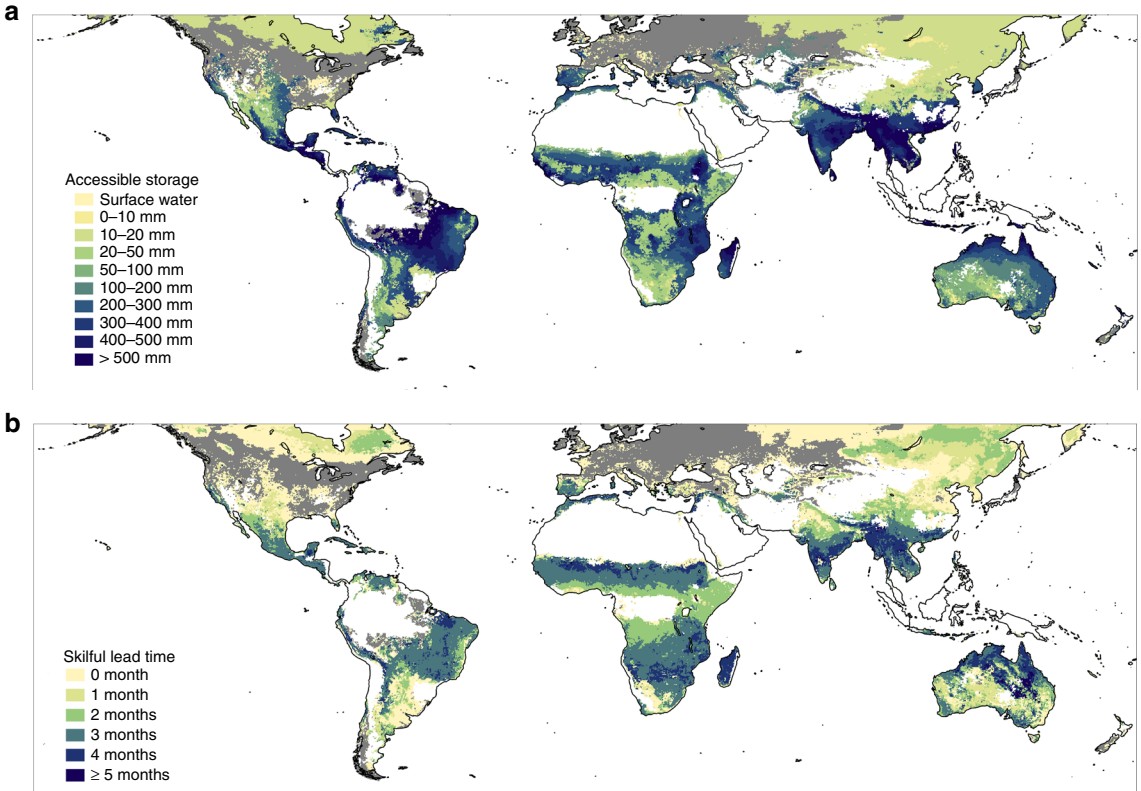

**Fig. 2** Maximum accessible storage capacity and skilful forecast lead time. **a** Accessible storage here relates to the soil depth to which vegetation Normalised Difference Vegetation Index (NDVI) responds most strongly. **b** Lead time for skilful vegetation condition forecasts. Lead time is counted from current month (0) to over 5 months. The 0-month lead time implies that skilful greenness predictions can only be made for the current month. Unvegetated and wet regions were masked out in white as Fig. 1a. The areas where vegetation are less responsive to water are shaded in grey

The assimilation of satellite-observed water dynamics into an ecohydrological model enables the estimation of vegetation-accessible storage, providing insights into dryland ecology as well as providing a basis for seasonal drought impact forecasting. Knowing how vegetation accesses water below the surface illuminates potential vegetation condition in dry environments and their buffering capacity to mitigate against droughts of different duration and intensity. This in turn can inform effective action to prepare and manage for drought.

## Methods

**Study area**. We limited the study region to include only arid to moderately humid vegetated land, defined by a dryness index of >0.3. We defined dryness as the average fraction of months that the mean potential evapotranspiration exceeds mean precipitation. The potential evapotranspiration was calculated using the PenmanMonteith equation[33] with 30 years of meteorological data[34,35]. Greenness was derived from the MODIS MOD13C2 NDVI product (https://lpdaac.usgs.gov/), which is a monthly composite of cloud-free observations resampled globally to 0.05° resolution. We regarded areas with maximum NDVI <0.25 through time as unvegetated and excluded them from our analysis. Our study region covered about 50% of total land area and 90% of the vegetated area.

**Ecohydrological model**. The World-Wide Water (W3) model[11] (http://wald.anu.edu.au/) simulates water stores and flows in vegetation, surface water, soil and unconfined groundwater systems. The model was driven by global estimates of daily precipitation[34], radiation, air temperature, wind speed, snowfall rate and surface pressure[35]. Soil and vegetation water and energy fluxes were simulated independently for deep-rooted vegetation and shallow-rooted vegetation in each hydrological response unit with different aerodynamic control of evaporation and interception capacities. The soil water store was separated into three unsaturated soil layers, namely, top (0–5 cm), shallow (5–100 cm) and deep (1–10 m) layer, where shallow-rooted vegetation and deep-rooted vegetation have different degrees

of access to moisture in the different soil layers. The unconfined groundwater store was estimated with the mass balance from the groundwater storage, deep drainage from deep soil layer, capillary rise from the groundwater, groundwater evaporation and groundwater discharge. The W3 model also includes the simulation of canopy and biomass change coupling with water balance dynamics. The water in the biomass, surface water, soil and groundwater comprised the total water storage in the W3 model.

**Data assimilation**. Three contrasting satellite water observations with different penetration depths from surface to the total water column were used in this study, namely, surface water extent, near-surface soil moisture and changes in total water storage. The surface water extent was estimated from MODIS 8-day composites using the reflectance dissimilarity between water and dry surfaces in shortwave infrared spectral band[18], analogous to the microwave method of estimating water extent using brightness temperature[36]. The MODIS-derived surface water extent was assimilated into the W3 model through a simple nudging approach with a high gain from the MODIS water fraction estimations to describe surface water dynamics not reliably simulated by the model. Monthly 3° × 3° GRACE mascon solutions[37] were obtained from the Jet Propulsion Laboratory (http://grace.jpl.nasa.gov). In contrast to GRACE, which has the capability of detecting water storage change accumulated in the total water column, SMOS can only penetrate the land surface for up to 5 cm. The 0.25° × 0.25° retrievals of near-surface soil moisture from the Centre Aval de Traitement des Données SMOS[38] (https://www.catds.fr) for both ascending and descending orbits were used to derive the daily averaged soil moisture content and to constrain the model simulated top-layer soil moisture (0–5 cm). To resolve the disparity in spatial, vertical and temporal resolution, the GRACE and SMOS data were assimilated into the W3 model using an Ensemble Kalman Smoother with a fixed 1-month window[21]. A single monthly GRACE observation together with all the daily SMOS observations within a 1-month window were included in the observation vector. The state vector was comprised of all model estimates of daily soil water storage in three layers and groundwater over a month and updated with GRACE and SMOS simultaneously. The observation operator including temporal accumulation components enables direct comparison with the GRACE and SMOS observations. The forecasts of water storage in different layers were adjusted with the Kalman gain matrix[39] based on the

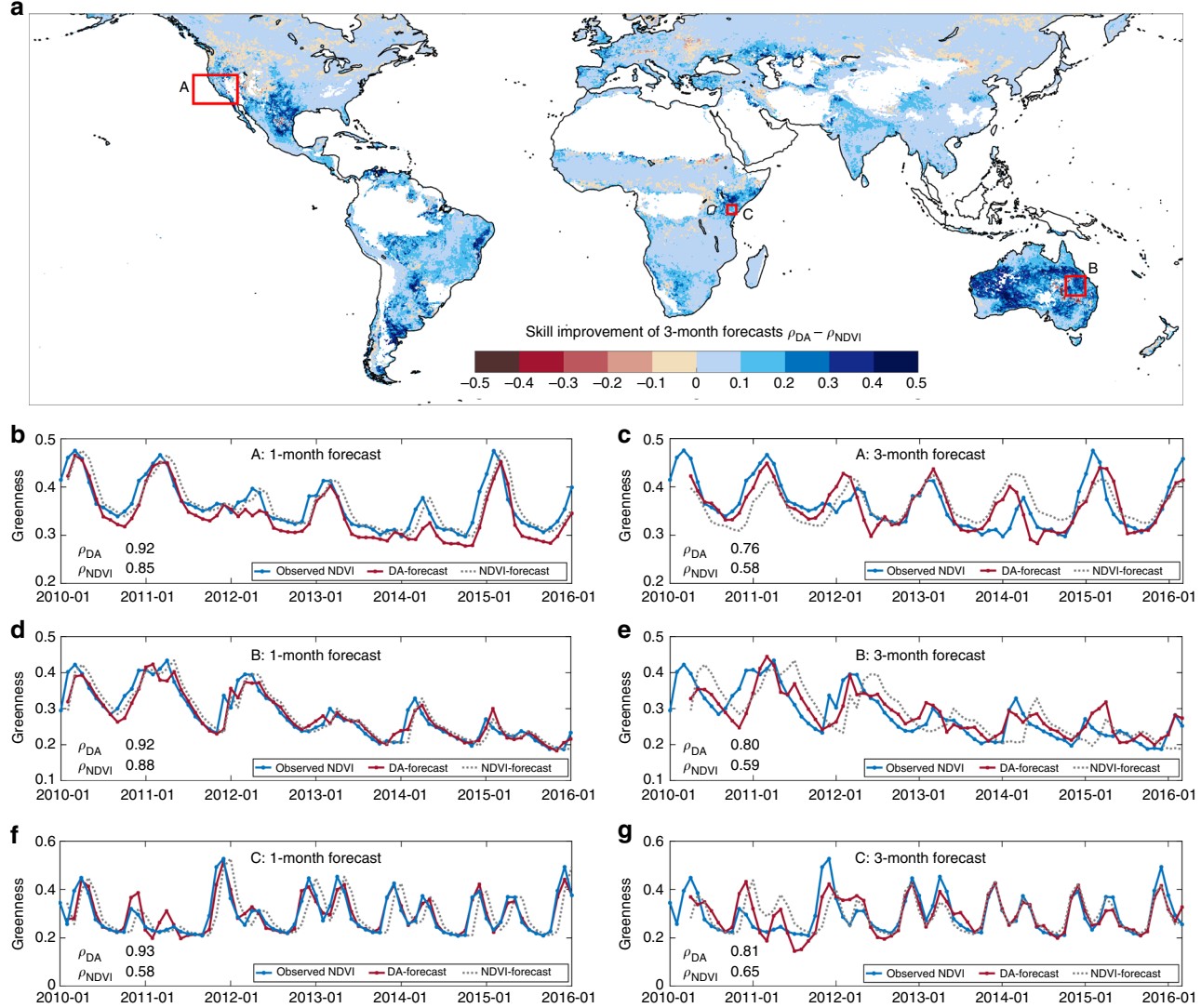

**Fig. 3** The 1-month and 3-month forecasts of vegetation condition. **a** Difference in correlation ($\rho$) between 3-month forecasts using accessible storage (DA-forecast, $\rho_{DA}$) and climatology (NDVI-forecast, $\rho_{NDVI}$) with greenness observations from 2010 to 2016. (DA: data assimilation, NDVI: Normalised Difference Vegetation Index). **b**–**g** Monthly time series of averaged 1-month and 3-months forecasts of greenness, compared with observed vegetation greenness over regions A, B and C in **a**

uncertainties in the W3 model and satellite observations. The model uncertainties were estimated from the sample covariance computed from 100 ensemble members which were generated through the perturbation of meteorological forcings (precipitation, air temperature and radiation in this case). The observation uncertainties were quantified using the spatially and temporally varying uncertainties in the GRACE and SMOS products. GRACE and SMOS observations imparted different constraints on the estimation of water storage at different layers through both model physics and simultaneous adjustment from variance–covariance structure between model states and observations. The smoother approach with a 1-month assimilation window also considered the temporal correlation between model states to separate water storage change into different depths based on different temporal dynamics. Data assimilation produced daily global 0.25° × 0.25° estimates of water in the near-surface soil, shallow root zone, deep root zone and unconfined groundwater.

**Statistical forecasts.** The statistical relationships between water storage dynamics and vegetation greenness anomalies were assessed using Spearman's rank correlation ($\rho$). The lagged $\rho$ between water storage integrated over different depths and greenness anomalies over the subsequent 1 to 12 months was calculated and used to define an optimal integration depth (in mm of equivalent water thickness), interpreted as the vegetation-accessible storage. Given accessible storage as a time-dependent variable, the 98th percentile of the

accessible storage over the study period at each grid was calculated as the maximum storage for the soil layer that vegetation growth responds to most strongly. The number of months for which lagged $\rho > 0.6$ was used as an indicator of skilful forecast lead time. The specific value of threshold used was based on maximising skilful forecasts. Nevertheless, the area of skilful forecasts remains stable with changes in threshold values. Alternative predictors tested included an antecedent precipitation index with a constant decay coefficient of 0.9[40], the satellite-derived SMOS soil moisture, GRACE total column storage estimates and the water storage estimates from model open-loop run without any data assimilation.

A deterministic forecast of the vegetation greenness anomaly $dV_t$ in $t$ month's time was obtained from a linear combination of the current greenness anomaly $dV_{t_0}$ and the anomaly in water storage over the determined optimal integration depth $z$, denoted by $S_{z,t_0}$ as follows:

$$dV_t = dV_{t_0} + \beta_1 S_{z,t_0} + \beta_2, \tag{1}$$

where $\beta_1$ and $\beta_2$ are regression coefficients. Comparison was made with persistence forecasts and climatology forecasts. The persistence forecast simply assumes the next month having the same anomaly as current month, $dV_t = dV_{t_0}$. Climatology forecasts use the average of previous available observations for month $t$ as the forecasts. The study period was limited to 6 years by the available observations and forcing data, starting from the launch of SMOS in 2010 to the end of the forcing data archives at the end of 2015.

Independent hindcast evaluation was achieved by splitting the time series into three equal segments; the performance for each time segment was calculated using a forecast model derived from data for the other two time segments. The averaged seasonal cycle excluding the evaluation period was added to the predicted greenness anomalies to obtain absolute greenness. The skill of water storage-derived forecasts was evaluated against the best performance from two NDVI-based forecasts at each pixel.

## Data availability

The World-wide water (W3) model is available online at http://wald.anu.edu.au. JPL GRACE land mascon solutions are available at http://grace.jpl.nasa.gov, supported by the NASA MEaSUREs Program. The CATDS level-3 daily soil moisture retrievals can be access through sipad (https://www.catds.fr/sipad/). The MOD13C2 NDVI data were retrieved from online Data Pool, courtesy of the NASA EOSDIS Land Processes Distributed Active Archive Center (LP DAAC), USGS/Earth Resources Observation and Science (EROS) Center, Sioux Falls, South Dakota, https://lpdaac.usgs.gov. The WFDEI meteorological forcing data can be retrieved from http://www.eu-watch.org/data_availability. Access to the MSWEP precipitation dataset is via http://www.gloh2o.org.

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

## Acknowledgements

This research was supported through ARC Discovery grant DP140103679. We thank Professor Michael L. Roderick and Professor Jeffery P. Walker for their kind help and suggestions in data analysis. This research was undertaken with the assistance of resources and services from the National Computational Infrastructure (NCI), which is supported by the Australian Government.

## Author contributions

All authors contributed to the development of the paper. S.T., A.I.J.M.v.D., P.T. and L.J.R. jointly designed this study. S.T. and A.I.J.M.v.D. prepared the dataset and S.T. conducted the analysis. A.I.J.M.v.D., P.T. and L.J.R. supervised the analysis. S.T and A.I.J.M.v.D. drafted the first manuscript. All authors contributed to the interpretation of the results and the drafting of the paper.

## Additional information

**Competing interests:** The authors declare no competing interests.

