## [Peer Review File · Nature Communications]

Reviewers' comments:

Reviewer #1 (Remarks to the Author):

The authors present a simple model based on current anomalies in the vegetation index NDVI and water storage computed over an optimized integration depth (Line 149), that they called accessible water storage, to forecast anomalies with respect to the seasonal cycle of NDVI in dry regions with a few months of advance. The authors show that the presented method has better forecast skills than using antecedent precipitation or remotely sensed near surface soil-moisture (SMOS) or total water storage (GRACE) products (Fig. 1d). Improvements with respect of using NDVI climatology are somewhat lower but significant in certain regions (Fig. 3). The authors concluded that such results shed light on how vegetation access water below the surface and inform specific actions on how to prepare and manage the onset of droughts (Line 111-115). While I might agree on the statistical improvement of forecasting skills in NDVI anomalies in comparison to other methods, the practical implications of such an improvement in comparison to, for instance, NDVI seasonality on the management of droughts are much less clear to me from the very succinct discussion. Are differences seen in the time series in Figure 3, so much important? If yes, why? If a skillful lead time is defined with a different (higher) threshold of correlation (Line 59-61) are results changing? How robust are they to this definition?

Additionally, I disagree with authors in defining the variable they use as "accessible storage". Admittedly, how this variable is derived is not very clear in the method section (see my specific comments below). However, according to lines 138 and the presented equation at line 149, this water storage is computed in some "optimal integration depth" that allows to maximize the correlation in vegetation anomalies in a given calibration period. This might have very little to do with the real "accessible water storage", as well as with how vegetation accesses water belowground, which may explain values of accessible water of only 50 mm or less for humid areas (Line 67), a clearly debatable value. Moreover this variable should be time dependent because of changes in soil moisture (and it is written as such in Line 149) but it is reported as static in Figure 2 or when referred in the text. I am not sure if/where the constraints in soil textural properties and rooting depth have been used in defining "vegetation accessible water storage". In other words, this is likely just a variable of the presented forecast model. This points does not detract from the forecast skill, but it undermines the link with the physical process.

Most important, the fact that in semi-arid and arid locations future anomalies in NDVI should be strongly influenced by the current profile of soil moisture is very much expected, as recognized by the authors themselves (Line 12-14). Allowing one "calibration parameter", the accessible water storage, in a simple model is also likely to improve forecast in comparison to other methods. Therefore, I am not sure I can highlight what I am really learning from this article, beyond the practicality of the forecasting method. Maybe, a better and longer explanation of the methodology and of the data-assimilation in the model would have helped, but once the physical link with "accessible water storage" is missing, the overall message is rather technical.

Specific comments

Line 19-20. I do not think, this sentence is very clear, especially at the abstract level.

Line 43. I would suggest to refer to "global – scale" or "large – scale" hydrological models, at the hillslope or profile scale vertical soil moisture distribution can be definitely captured much better.

Line 62. Atmospheric water demand is typically the "vapor pressure deficit", here, instead you are referring to potential evapotranspiration (as stated in Line 119), I would write it explicitly.

Line 67-70. Accessible water storage is a dynamic variable. Are these values “averages”, “maximum values”? Generally, how this is computed should be explicit in the text? 50 mm of accessible water storage sounds as a very small quantity. If I look for instance to North Carolina/Georgia, forested areas have definitely roots at least 1 m deep that with a sandy-loam texture will make roughly 300 mm of available storage at saturation and 150 mm at field capacity. Figure 2 reports values of 10-20 mm.

Line 77. Beyond Figure 1d, I would also show some Figure with time series to better illustrate the comparison of your method with alternative methods.

Line 50 and 109. How satellite observed water dynamics are integrated in the ecohydrological model and what the ecohydrological model does remain obscure throughout the entire paper and also in the method section.

Line 119. Where is potential evapotranspiration taken from? How is it computed?

Line 124. How is possible that you cover 90% of vegetated land are if you limit the study to region with dryness index >0.3 (Line 117-118)? I think you refer to $NDVI < 0.25$ mask, but this is not very clear.

Line 128-135. How all these products are exactly used in the data-assimilation framework remains unclear. Does the model have a vertical explicit representation of the soil profile? How many layers? How groundwater is represented? What is the role of surface water? Until which depth the SMOS product is used? How the GRACE product anomalies are integrated with the other products? Does the W3 model accounts for soil textural properties and rooting depth? How is this linked to the “optimal integration depth” computed at Line 149? Generally, the methodological description should be much more accurate and much more extensive.

Reviewer #2 (Remarks to the Author):

I find this paper very interesting

I have a few suggestions to make this paper a bit better

[1] A few more details in methods - how is the model constructed? The satellite estimates - SMOS and GRACE cannot estimate the 500mm soil moisture - how is this done?

[2] In Figure 3 a few statistics on time series between the DA and NDVI forecast and the degradation between 1 and 3 months would be useful

Overall, a very interesting paper

Reviewer #3 (Remarks to the Author):

With great interest I have read the manuscript “Forecasting dryland condition months in advance through satellite data assimilation”. The starting point of the study is the importance of having a reliable knowledge of the water stored in the earth system: Authors claim (and this is supported by literature) that knowledge of the water stored in the system contributes more to predictability than the climate system state. An improved knowledge of this water stored in the system would then lead to an improved predictability. Then, the main objective of the study is to improve the skill of forecasting vegetation condition months in advances (and in that respect I believe that ‘vegetation’ is a key word to this study and deserves to be in the title). To that end initializing a forecast using improved initial condition, i.e. an analyzed state of the land surface variables is foreseen. Authors propose a system where different water-related satellite-derived information, namely MODIS surface water extent, SMOS near surface soil moisture and GRACE total column

water storage are used/assimilated in the World-Wide Water (W3) model using an Ensemble Kalman Smoother (for SMOS and GRACE data). Satellite-derived Normalized Difference Vegetation Index (NDVI) anomalies are compared to anomalies in water storage from Author's system to develop a simple forecast model.

The manuscript is clearly of interest of the scientific community, well written in very good English (although I am not a native English speaker) and references are appropriate. I believe a "communication" paper needs a balanced complexity/explanations ratio. Some parts of the manuscripts remain unclear, at least to me, some details are required before it can deserve publication (of course, Authors may want to prove me wrong). Please see below an attempt to help.

L.39, Soil moisture indeed has a memory and likely vegetation has one, too. I am missing something here, could you (even briefly) comment on how vegetation (flows in vegetation) is represented in your system? I am wondering if the vegetation part of your system (then of the study) may not be accurate enough to lead to proposed perspectives (agricultural decisions, drought relief...).

L.48-50, OK for SMOS and GRACE but how surface water extent are used/assimilated? The Methods section only mention the assimilation of SMOS and GRACE (?). Assuming that GRACE represents the total column water storage, how its assimilation competes with the one of SMOS and with the use of MODIS surface water extent. Could you comment on assimilating GRACE data only (link to figure 1d)? I understand from lines 128-129 that the MODIS data are used to compensate model deficiency describing water dynamics (?) could you please add how?

L.141-145, what was the main outcome if those test? (i.e. could you comment more on lines 75-78, in line with my previous comment). I have the feeling that caption of figure 1d tries to answer this but it is still unclear to me. (Second (b) panel of figure 1 should be (c)).

Please define symbol in figure 3

Response to reviewers

We would like to thank the Editor and reviewers for the thoughtful comments. We have revised the manuscript to address the reviewers' concerns and have provided further clarification of the method and results. We believe the revised manuscript addresses the issues raised and, as a result has improved significantly. In the revised manuscript, we:

- include more details of the model structure and data assimilation approach.
- write a separate section discussing the results and comparing the accessible storage with existing studies.
- provide three supporting figures in the response to address reviewers' concerns
- modified Figure 2 by masking out the areas with low forecast skills.

The supporting figures can be included in the supplementary materials if the Editor deems that useful.

Below we respond (in **blue**) to each of the comments (in **black**) in detail. The relevant line numbers and figures (in **red**) referring to the revised manuscript were given where necessary. A revised manuscript with tracked changes (**highlighted in yellow**) is included in the submission entitled as the "Related Manuscript File".

Reviewers' comments

Reviewer #1 (Remarks to the Author):

The authors present a simple model based on current anomalies in the vegetation index NDVI and water storage computed over an optimized integration depth (Line 149), that they called accessible water storage, to forecast anomalies with respect to the seasonal cycle of NDVI in dry regions with a few months of advance. The authors show that the presented method has better forecast skills than using antecedent precipitation or remotely sensed near surface soil-moisture (SMOS) or total water storage (GRACE) products (Fig. 1d). Improvements with respect of using NDVI climatology are somewhat lower but significant in certain regions (Fig. 3). The authors concluded that such results shed light on how vegetation access water below the surface and inform specific actions on how to prepare and manage the onset of droughts (Line 111-115).

While I might agree on the statistical improvement of forecasting skills in NDVI anomalies in comparison to other methods, the practical implications of such an improvement in comparison to, for instance, NDVI seasonality on the management of droughts are much less clear to me from the very succinct discussion. Are differences seen in the time series in Figure 3, so much important? If yes, why?

We agree with reviewer that a statistical improvement of forecasting skill is not automatically practically important, especially when the resulting skill is still relatively low (e.g. from 0.2 to 0.4). However, we believe that an increase of 0.2 in correlation as shown in Figure 3 (e.g. from 0.6 to 0.8) is a substantial improvement and can certainly make a difference in aspects of drought management (Line 26-31). To describe three practical examples we are familiar with, knowing three months in advance of potential vegetation drought impacts, famine relief organisations can start planning logistics, fire agencies can start preparing for increased fire risk, and graziers can move or sell their stock, to name just

a few. In the revised manuscript, we added a new section of Discussion (from Line 123 to 150). In particular, we included the discussion about the importance of our results in Line 137 – 143.

“Our findings suggest that incorporating current soil water availability, specific to different vegetation type and aridity, can significantly improve the accuracy of vegetation condition forecasts 3 months in advance for the majority of the drylands globally. Such forecasts can help to improve drought early warning system and reduce economic and environmental damage. This capacity may become even more important in the context of projected increases in the occurrence and severity of drought under climate change in some regions³⁰⁻³²”

If a skillful lead time is defined with a different (higher) threshold of correlation (Line 59-61) are results changing? How robust are they to this definition?

We appreciate the reviewer’s concern. We also calculated the skillful lead time using thresholds of 0.7 and 0.8 and compared them with results for a threshold of 0.6 as shown in the following figure. The area with skillful forecast is maintained for majority of the dryland area.

In addition, we also calculated the fraction of the residuals of NDVI climatology forecasts and forecasts using accessible storage as $\frac{1-\rho_{ndvi}}{1-\rho_{da}}$. A value larger than 1 means that our forecasts based on accessible storage are more skillful than climatology forecasts, and a value of 2 can be interpreted as meaning they are 2 times more skillful in terms of residual unexplained variance. As shown in the figure below, forecasts derived from accessible storage after the assimilation increased the accuracy of the forecasts by more than 2 or 3 times against climatology forecast for a large part of the global dryland area. Since our results remain robust, we will not include the supporting figures in the manuscript but those figures can be included in the supplementary materials if editor deems that useful. Areas with low forecast skill were shaded here in grey to be consistent with the revised Figure 2 in the manuscript (see response further below).

Additionally, I disagree with authors in defining the variable they use as “accessible storage”. Admittedly, how this variable is derived is not very clear in the method section (see my specific comments below). However, according to lines 138 and the presented equation at line 149, this water storage is computed in some “optimal integration depth” that allows to maximize the correlation in vegetation anomalies in a given calibration period. This might have very little to do with the real “accessible water storage”, as well as with how vegetation accesses water belowground, which may explain values of accessible water of only 50 mm or less for humid areas (Line 67), a clearly debatable value.

Agreed. We thank the reviewer for the opportunity to clarify the definition of “accessible storage” and we have tightened our terminology in response. The accessible storage is not necessarily the entire total root-zone soil water capacity. Rather, it is the water store that empirically drives observed vegetation anomalies. That is, it is the soil water store that can explain the observed vegetation patterns at large scale, and is likely readily accessible by the dominant vegetation component. Although some plants may well access deeper soil water than this inferred storage, the largescale response observed indicates that the ecosystem as a whole responds to the shallower soil water availability defined by our results. We have added some explanatory text in Lines 126 – 132.

“The accessible storage capacity inferred here is empirically defined and may be less than the total moisture storage that can be accessed by the deepest-rooted individuals within the ecosystem. Rather, our results indicate the soil water store that empirically best predicts vegetation anomalies for the visually dominant ecosystem component as observed by remote sensing. Nonetheless, in semi-arid to arid regions we found spatial patterns that are very similar to previously reported root-zone storage capacity and rooting depths^{15,24}.”

Moreover this variable should be time dependent because of changes in soil moisture (and it is written as such in Line 149) but it is reported as static in Figure 2 or when referred in the text. I am not sure if/where the constraints in soil textural properties and rooting depth have been used in defining “vegetation accessible water storage”. In other words, this is likely just a variable of the presented forecast model. This point does not detract from the forecast skill, but it undermines the link with the physical process.

Accessible storage is indeed a time dependent variable. We clarified that the maximum values were shown in Figure 2 in both figure caption and the text (Line 216).

“Given accessible storage as a time dependent variable, the 98th percentile of the accessible storage over the study period at each grid was calculated as the maximum storage for the soil layer that vegetation growth responds most strongly to.”

More details of the model are now provided in the Method section in the revised manuscript (Line 165-176). Neither soil properties nor rooting depth information were used in the calculation of accessible storage. The accessible storage was empirically calculated from the soil water storage in different layers estimated from the ecohydrological model (W3) constrained by satellite observations through data assimilation. Therefore, the accessible storage is strongly linked to physical processes, but in an empirical sense.

Conversely, we would argue that in fact ‘rooting depth’ is a term that sounds straightforward in theory, but that is in fact very hard to define in any useable way in practice. Firstly, different individuals within the ecosystem have different maximum rooting depth. Secondly, the ability for individual plants to extract moisture from deeper layers typically declines with depth, meaning deeper layers are less ‘accessible’ even though the deepest root may technically reach there. Our approach to determining accessible storage is empirical and therefore returns an ‘effective’ water storage depth that shows the strongest link, which can still be interpreted in terms of physical processes, and provides useful insights about it (Line 126 - 132).

Most important, the fact that in semi-arid and arid locations future anomalies in NDVI should be strongly influenced by the current profile of soil moisture is very much expected, as recognized by the authors themselves (Line 12-14). Allowing one “calibration parameter”, the accessible water storage, in a simple model is also likely to improve forecast in comparison to other methods. Therefore, I am not sure I can highlight what I am really learning from this article, beyond the practicality of the forecasting method. Maybe, a better and longer explanation of the methodology and of the data-assimilation in the model would have helped, but once the physical link with “accessible water storage” is missing, the overall message is rather technical.

We agree with the reviewer that soil moisture is the controlling factor for vegetation in arid region. This was, of course, the motivation for our use of soil moisture profile information to forecast vegetation condition.

Accessible storage is not a “calibration parameter”, it is a time dependent variable including the water storage at a specified integration depth. In our case we used a fixed integration depths in the forecast of NDVI at different lead time (1-month, 2-month, 3month...), and there was no calibration of integration depths based on different lead time.

We have added more detail to the explanation of methodology as well as the model physics in the Method section (Line 165-176). Responses to the reviewer’s specific comments can be found below.

Specific comments

Line 19-20. I do not think, this sentence is very clear, especially at the abstract level.

Agreed. We removed this sentence from the abstract. We have clarified this sentence in the Discussion section (Lines 126-131).

“The accessible storage capacity inferred here is empirically defined and may be less than the total moisture storage that can be accessed by the deepest-rooted individuals within the ecosystem. Rather, our results indicate the soil water store that empirically best predicts vegetation anomalies for the visually dominant ecosystem component as observed by remote sensing.”

Line 43. I would suggest to refer to “global – scale” or “large – scale” hydrological models, at the hillslope or profile scale vertical soil moisture distribution can be definitely captured much better.

Agreed. Changed to *“This suggests the potential to use root-zone soil water availability to forecast vegetation condition at large scale.”* In Line 40.

Line 62. Atmospheric water demand is typically the “vapor pressure deficit”, here, instead you are referring to potential evapotranspiration (as stated in Line 119), I would write it explicitly.

Agreed. Changed to *“The results were analysed as a function of climate dryness at each location, defined as the long-term average number of months for which potential evapotranspiration exceeds precipitation (see Methods).”* in Line 70.

Line 67-70. Accessible water storage is a dynamic variable. Are these values “averages”, “maximum values”? Generally, how this is computed should be explicit in the text?

They represent maximum values. We clarified in both the figure caption and the text. Specifically we have added:

“Given accessible storage as a time dependent variable, the 98th percentile of the accessible storage over the study period at each grid was calculated as the maximum storage for the soil layer that vegetation growth responds most strongly to.” in Line 216.

50 mm of accessible water storage sounds as a very small quantity. If I look for instance to North Carolina/Georgia, forested areas have definitely roots at least 1 m deep that with a sandy-loam texture will make roughly 300 mm of available storage at saturation and 150 mm at field capacity. Figure 2 reports values of 10-20 mm.

As per our earlier response, our vegetation-accessible storage is not equal to root-zone water capacity, but the water store that empirically controls vegetation growth anomalies. Our accessible storage shows similar spatial patterns with the rooting depths and root-zone

water capacity over dryland areas, but we agree that its interpretation in climates that are not primarily water-limited is more complicated. North Carolina/Georgia is such an environment, experiencing a warm temperate and moist climate where natural forest vegetation can be assumed to respond mainly to radiation and temperature (Nemani et al, 2003; Wu et al, 2015). The correlation between water availability and vegetation greenness is relatively low in this and most other areas where water is not the dominant growth factor, and we agree that our results need to be interpreted with some scepticism in such cases. To avoid this confusion, we have masked out areas where the correlation between vegetation conditions and soil water availability is low ($r < 0.2$, mainly in North America and Europe) in the revised Figure 2, as shown below.

More discussion of our results compared to other studies was added in the revised manuscript:

“Nonetheless, in semi-arid to arid regions we found spatial patterns that are very similar to previously reported root-zone storage capacity and rooting depths^{15,24}.” in Line 130-132.

Line 77. Beyond Figure 1d, I would also show some Figure with time series to better illustrate the comparison of your method with alternative methods.

Below we include time series plots of the 3-months NDVI forecasts using GRACE, SMOS, API data for the same regions in Figure 3. The 3 months forecasts with accessible storage perform the best among the alternative methods with a higher correlation (e.g. increase from 0.33 for API-forecast to 0.81 for DA-forecast). Since GRACE data have missing values two or three time per year after 2005, it causes many missing values and uncertain regression results affect the forecasts.

Since the emphasis of this study was to demonstrate the improvements of NDVI forecasts using accurate soil water availability compared to climatology forecasts, we think providing these additional figures of the time series in the manuscript is more distracting than helpful. We have demonstrated the benefits of data assimilation in our previous published paper (Tian et al, 2017). A summary figure as Figure 1d is sufficient to demonstrate the skills of assimilation, and we would like to keep the time series plots with the independent comparison of forecasts using climatology as shown in Figure 3. However, we would be happy to provide the time series plots below in the Supplementary Material, if the Editor and reviewer think it would be useful.

Line 50 and 109. How satellite observed water dynamics are integrated in the ecohydrological model and what the ecohydrological model does remain obscure throughout the entire paper and also in the method section.

We hope it is clearer in the revised manuscript. We added the details of satellite data integration in the Method section as shown in the highlighted text from Line 194-209.

“To resolve the disparity in spatial, vertical and temporal resolution, the GRACE and SMOS data were assimilated into the W3 model using an Ensemble Kalman Smoother with a fixed one-month window²². The state vector was comprised of all model estimates of daily soil water storage in three layers and groundwater over a month and updated with GRACE and SMOS simultaneously. The observation operator including temporal accumulation components, enables direct comparison with the GRACE and SMOS observations. The forecast of water storage at different layers, were adjusted with the Kalman gain matrix⁴⁰ based on the uncertainties in the W3 model and satellite observations. The model uncertainties were estimated from the sample covariance computed from 100 ensemble members which were generated through the perturbation of meteorological forcings (precipitation, air temperature and radiation in this case). The observation uncertainties were quantified using the spatially and temporally varying uncertainties in the GRACE and SMOS products. GRACE and SMOS observations imparted different constraints on the estimation of water storage at different layer through both model physics and simultaneous adjustment from variance-covariance structure between model states and observations. The smoother approach with one-month assimilation window also considered the temporal correlation between model states to separate water storage change into different depths based on different temporal dynamics.”

Line 119. Where is potential evapotranspiration taken from? How is it computed?

We clarified this in the Method section as follow:

“The potential evapotranspiration was calculated using the Penman-Monteith equation³³ with 30 years of meteorological data^{34,35}.” in **Line 155**

Line 124. How is possible that you cover 90% of vegetated land area if you limit the study to region with dryness index >0.3 (Line 117-118)? I think you refer to NDVI<0.25 mask, but this is not very clear.

The size of study area is in total 187,066 grid cells. The vegetated land area (203,038 grid cells) refers to the area masked out with the requirement that maximum NDVI is less than 0.25. We clarified this in the Method section:

“We regard areas with maximum NDVI < 0.25 through time as unvegetated and excluded them from our analysis. Our study region covered about 50% of total land area and 90% of the vegetated area.” in **Line 158**

Line 128-135. How all these products are exactly used in the data-assimilation framework remains unclear.

We have added explanatory text about the model structure, datasets and data assimilation method in more details in the Method section in the revised manuscript (**Line 166-211**). Specific concerns are addressed below.

Does the model have a vertical explicit representation of the soil profile? How many layers?

We have now added:

“Soil and vegetation water and energy fluxes were simulated independently for deep-rooted vegetation and shallow-rooted vegetation in each hydrological response unit (HRU) with different aerodynamic control of evaporation and interception capacities. The soil water store was separated into three unsaturated soil layers, namely, top (0-5cm), shallow (5-100cm), and deep (1-10m) layer, where shallow-rooted vegetation and deep-rooted vegetation have different degrees of access to moisture in the different soil layers.” in Line 165

How groundwater is represented? What is the role of surface water?

We have clarified the groundwater and surface water representation in the revised manuscript as follow:

“The unconfined groundwater store was estimated with the mass balance from the groundwater storage, deep drainage from deep soil later, capillary rise from the groundwater, groundwater evaporation and groundwater discharge.” in Line 171

“The surface water extent was estimated from MODIS 8-day composites using the reflectance dissimilarity between water and dry surfaces in shortwave infrared spectral band¹⁹, analogous to the microwave method of estimating water extent using brightness temperature³⁷. The MODIS-derived surface water extent was assimilated into the W3 model through a simple nudging approach with a high gain from the MODIS water fraction estimations to describe surface water dynamics not reliably simulated by the model.” in Line 179

Until which depth the SMOS product is used?

We added clarification:

“The 0.25 °×0.25 ° retrievals of near-surface soil moisture from the Centre Aval de Traitement des Données SMOS39 (<https://www.catds.fr>) for both ascending and descending orbits were used to derive the daily averaged soil moisture content and constrain the model simulated top-layer soil moisture (0-5cm). ” in Line 189.

“The state vector was comprised of all model estimates of daily soil water storage in three layers and groundwater over a month and updated with GRACE and SMOS simultaneously.” in Line 194.

“GRACE and SMOS observations imparted different constraints on the estimation of water storage at different layer through both model physics and simultaneous adjustment from variance-covariance structure between model states and observations.” in Line 205.

How the GRACE product anomalies are integrated with the other products?

This was described in detail in Tian et al. (2017), but we have added more details in the revised Method section from Line 177 to Line 210.

Does the W3 model accounts for soil textural properties and rooting depth? How is this linked to the “optimal integration depth” computed at Line 149?

Unlike many land surface models, W3 deliberately simulates only soil layers with a nominal water storage capacity instead of prescribing a layer thickness and porosity. The parameters regard to soil layer specification are the field capacity values for three layers. In this work, the accessible storage or the optimal integration depths are water storage in water thickness (mm) instead of physical depths.

“The lagged ρ between water storage integrated over different depths and greenness anomalies over the subsequent 1 to 12 months was calculated and used to define an optimal integration depth (in water thickness, i.e. mm), interpreted as the vegetation-accessible storage.” in Line 217

Generally, the methodological description should be much more accurate and much more extensive.

In the revised manuscript, we added more information about the assimilation and NDVI forecast methods, as shown in the highlighted text (Line 152-220).

Reviewer #2 (Remarks to the Author):

I find this paper very interesting

I have a few suggestions to make this paper a bit better

[1] A few more details in methods - how is the model constructed?

Thank you. In response to both reviewers, we have included more details about the model structure in the Method section (Line 166-176), including the details of water balance simulation in soil water and groundwater.

The satellite estimates - SMOS and GRACE cannot estimate the 500mm soil moisture - how is this done?

We understand the potential confusion. We added the measurement depths and limitations of GRACE and SMOS in both introduction and method section. We also added the assimilation method in more details that explains the integration of SMOS and GRACE in estimating root-zone soil water storage.

“Satellite remote sensing has been pivotal to deepening our understanding of water availability and climate change at regional-to-global scale, and has helped to advance predictive models and decision making¹⁷. However, satellite observations of water presence are limited to the either surface (up to 5 cm for soil moisture, e.g. SMOS mission) or total water column (GRACE mission). The quantification of the vertical distribution of water

storage is extremely difficult over large spatial and time domains due to the lack of direct measurement of root-zone soil moisture and groundwater storage. The accuracy of soil moisture or groundwater storage estimates separated from total water storage is limited without ancillary data and the consideration of data uncertainties¹⁸.” in Line 48

“In contrast to GRACE, which has the capability of detecting water storage change accumulated in the total water column, SMOS can only penetrate the land surface for up to 5 cm. The 0.25°×0.25° retrievals of near-surface soil moisture from the Centre Aval de Traitement des Données SMOS³⁹ (<https://www.catds.fr>) for both ascending and descending orbits were used to derive the daily averaged soil moisture content and constrain the model simulated top-layer soil moisture (0-5cm). To resolve the disparity in spatial, vertical and temporal resolution, the GRACE and SMOS data were assimilated into the W3 model using an Ensemble Kalman Smoother with a fixed one-month window²². The state vector was comprised of all model estimates of daily soil water storage in three layers and groundwater over a month and updated with GRACE and SMOS simultaneously. The observation operator including temporal accumulation components, enables direct comparison with the GRACE and SMOS observations. The forecast of water storage at different layers, were adjusted with the Kalman gain matrix⁴⁰ based on the uncertainties in the W3 model and satellite observations. The model uncertainties were estimated from the sample covariance computed from 100 ensemble members which were generated through the perturbation of meteorological forcings (precipitation, air temperature and radiation in this case). The observation uncertainties were quantified using the spatially and temporally varying uncertainties in the GRACE and SMOS products. GRACE and SMOS observations imparted different constraints on the estimation of water storage at different layer through both model physics and simultaneous adjustment from variance-covariance structure between model states and observations. The smoother approach with one-month assimilation window also considered the temporal correlation between model states to separate water storage change into different depths based on different temporal dynamics.” in Line 187

[2] In Figure 3 a few statistics on time series between the DA and NDVI forecast and the degradation between 1 and 3 months would be useful.

Overall, a very interesting paper

We thank the reviewer for their suggestion and comment. In Figure 3, we have provided the correlation between the DA predicted NDVI and observed NDVI, together with the correlation between NDVI climatology predicted NDVI and observed NDVI. We also included more discussion on the statistics results in the revised manuscript in Line 116:

“Significant improvements with an increase in correlation for more than 0.2 were achieved with longer lead time compared with NDVI-climatology. Forecasts using accessible storage showed a slower decay of forecast skill than NDVI-based forecast by more than 0.1 units and maintained a correlation of ~0.8 in three-months forecast.”

Reviewer #3 (Remarks to the Author):

With great interest I have read the manuscript “Forecasting dryland condition months in advance through satellite data assimilation”. The starting point of the study is the

importance of having a reliable knowledge of the water stored in the earth system: Authors claim (and this is supported by literature) that knowledge of the water stored in the system contributes more to predictability than the climate system state. An improved knowledge of this water stored in the system would then lead to an improved predictability. Then, the main objective of the study is to improve the skill of forecasting vegetation condition months in advances (and in that respect I believe that 'vegetation' is a key word to this study and deserves to be in the title). To that end initializing a forecast using improved initial condition, i.e. an analyzed state of the land surface variables is foreseen. Authors propose a system where different water-related satellite-derived information, namely MODIS surface water extent, SMOS near surface soil moisture and GRACE total column water storage are used/assimilated in the World-Wide Water (W3) model using an Ensemble Kalman Smoother (for SMOS and GRACE data). Satellite-derived Normalized Difference Vegetation Index (NDVI) anomalies are compared to anomalies in water storage from Author's system to develop a simple forecast model.

The manuscript is clearly of interest of the scientific community, well written in very good English (although I am not a native English speaker) and references are appropriate. I believe a "communication" paper needs a balanced complexity/explanations ratio. Some parts of the manuscripts remain unclear, at least to me, some details are required before it can deserve publication (of course, Authors may want to prove me wrong). Please see below an attempt to help.

Thank you. We revised the title into "*Forecasting dryland **vegetation** condition months in advance through satellite data assimilation*" as the reviewer suggested. More details of the method and dataset were added in the Method section (Line 154-Line 241). Responses to the specific concerns are addressed below.

L.39, Soil moisture indeed has a memory and likely vegetation has one, too. I am missing something here, could you (even briefly) comment on how vegetation (flows in vegetation) is represented in your system? I am wondering if the vegetation part of your system (then of the study) may not be accurate enough to lead to proposed perspectives (agricultural decisions, drought relief...).

We have added a brief introduction of the model structure with more details of the separation of water store simulation with shallow-rooted and deep-rooted vegetation, as well as the coupling of biomass change with soil water availability (Line 166 - 176). The W3 model contains a simple vegetation phenology model that simulates changes in leaf biomass as a function of actual and ecohydrological equilibrium leaf biomass (itself determined by soil moisture availability and atmospheric demand), which in turn affects transpiration and soil hydrology. The NDVI forecast is based on statistical method using accessible storage derived from satellite data assimilation not estimated directly from W3 model. We hope this addresses the reviewer's question.

L.48-50, OK for SMOS and GRACE but how surface water extent are used/assimilated? The Methods section only mention the assimilation of SMOS and GRACE (?). Assuming that GRACE represents the total column water storage, how its assimilation competes with the one of SMOS and with the use of MODIS surface water extent. Could you comment on

assimilating GRACE data only (link to figure 1d)? I understand from lines 128-129 that the MODIS data are used to compensate model deficiency describing water dynamics (?) could you please add how?

The MODIS-derived surface water extent was assimilated into the model with a simple nudging approach which will not affect the other water stores. The SMOS and GRACE data were assimilated through the ensemble Kalman smoother to redistribute the soil water stores in three layers and groundwater through the variance-covariance matrix between states and observations. We have added more details of the assimilation method from Line 177-211:

“Three contrasting satellite water observations with different penetration depths from surface to the total water column were used in this study, namely, surface water extent, near-surface soil moisture and changes in total water storage. The surface water extent was estimated from MODIS 8-day composites using the reflectance dissimilarity between water and dry surfaces in shortwave infrared spectral band¹⁹, analogous to the microwave method of estimating water extent using brightness temperature³⁷. The MODIS-derived surface water extent was assimilated into the W3 model through a simple nudging approach with a high gain from the MODIS water fraction estimations to describe surface water dynamics not reliably simulated by the model. Monthly 3°×3° GRACE mascon solutions³⁸ were obtained from the Jet Propulsion Laboratory (<http://grace.jpl.nasa.gov>). In contrast to GRACE, which has the capability of detecting water storage change accumulated in the total water column, SMOS can only penetrate the land surface for up to. The 0.25°×0.25° retrievals of near-surface soil moisture from the Centre Aval de Traitement des Données SMOS³⁹ (<https://www.catds.fr>) for both ascending and descending orbits were used to derive the daily averaged soil moisture content and constrain the model simulated top-layer soil moisture (0-5cm). To resolve the disparity in spatial, vertical and temporal resolution, the GRACE and SMOS data were assimilated into the W3 model using an Ensemble Kalman Smoother with a fixed one-month window²². The state vector was comprised of all model estimates of daily soil water storage in three layers and groundwater over a month and updated with GRACE and SMOS simultaneously. The observation operator including temporal accumulation components, enables direct comparison with the GRACE and SMOS observations. The forecast of water storage at different layers, were adjusted with the Kalman gain matrix⁴⁰ based on the uncertainties in the W3 model and satellite observations. The model uncertainties were estimated from the sample covariance computed from 100 ensemble members which were generated through the perturbation of meteorological forcings (precipitation, air temperature and radiation in this case). The observation uncertainties were quantified using the spatially and temporally varying uncertainties in the GRACE and SMOS products. GRACE and SMOS observations imparted different constraints on the estimation of water storage at different layer through both model physics and simultaneous adjustment from variance-covariance structure between model states and observations. The smoother approach with one-month assimilation window also considered the temporal correlation between model states to separate water storage change into different depths based on different temporal dynamics.”

Figure 1d shows the forecasts using GRACE data in the forecasting not the results from the assimilation of GRACE data. This is to investigate which satellite water observation mainly

impacts on the results. The benefits of joint assimilation against the assimilation of GRACE data only were described by Tian et al (2017).

L.141-145, what was the main outcome if those test? (i.e. could you comment more on lines 75-78, in line with my previous comment). I have the feeling that caption of figure 1d tries to answer this but it is still unclear to me.

The main message of this test is (now in **Line 86**) to demonstrate that the data assimilation can improve the forecast skill with more accurate water availability information. Using satellite observations only or model only cannot achieve the accuracy that assimilation can obtain. Moreover, it shows that soil water availability is more important than precipitation in predicting vegetation conditions (API results). We added more discussion of this figure in the manuscript as shown below.

“Skilful forecasts using soil water availability from satellite observations or model simulations only could be achieved for no more than 20% of the vegetated arid area (dryness > 0.6).” in **Line 86**

“Our estimates of the accessible storage combine soil water dynamics information captured by multiple satellite sensors through data assimilation. A stronger response of vegetation greenness to water availability was found using accessible storage, when compared against water availability derived from only satellite observations or the ecohydrological model, including results from previous studies²⁶⁻²⁹” in **Line 133**

(Second (b) panel of figure 1 should be (c)).
Panel number fixed.

Please define symbol in figure 3

We have now defined the symbols used in figure3 in the caption.

Reference:

van Dijk, A.I.J.M. and Warren, G., 2010. The Australian water resources assessment system. *Version 0.5*, 3(5).

Tian, S. Y. *et al.* Improved water balance component estimates through joint assimilation of GRACE water storage and SMOS soil moisture retrievals. *Water Resources Research* **53**, 1820-1840, doi:10.1002/2016wr019641 (2017).

Wu, D. H. *et al.* Time-lag effects of global vegetation responses to climate change. *Global Change Biol* **21**, 3520-3531, doi:10.1111/gcb.12945 (2015).

Nemani, R. R. *et al.* Climate-driven increases in global terrestrial net primary production from 1982 to 1999. *Science* **300**, 1560-1563, doi:DOI 10.1126/science.1082750 (2003).

REVIEWERS' COMMENTS:

Reviewer #1 (Remarks to the Author):

The authors addressed the comments I made in the first article. Specifically they provided a more extensive and detailed explanation of what the model does, how datasets are assimilated, and how the "accessible water storage" is computed. These modifications improved significantly the manuscript.

I appreciate the explanation of methods that is now clearer in the text, but I still think "accessible water storage" is a misleading term because it is solely determined through NDVI anomalies and even though the authors mask regions where the model has low predictive skills, they cannot account for additional "accessible water storage" that does not result in a change in NDVI signal. This is written in the text, however, if somebody goes quickly to the article or worse stop to the results of Figure 2, he/she may interpret the reported value labeled as "accessible water storage" in a quite misleading way.

I might still have a quite different feeling on the practical relevance of the improved prediction skills of NDVI for real problems (now stated at Line 140-150) but I guess, here, we are entering the realms of opinions and the one of the authors is likely as valuable as mine, so I would leave it as it is.

Specific comments

Maybe you can add one sentence in the manuscript to say that the threshold chosen for the correlation does not affect much the results (Fig. 1 in the rebuttal). This is a point in favor of the study.

Line 147. I would suggest to not referring to "vegetation population dynamics". There is nothing in the article that is directly related to "population dynamics" and while water access surely affects plant competition, it is not this study that can shed light on how different plants access water and compete, what you retrieve it is just an estimated value of water storage over the grid area.

Line 173. "layers" rather than later.

Reviewer #3 (Remarks to the Author):

Dear Authors,

This is the second time I am reviewing the manuscript. I have now read your responses to my reviews as well as those from two anonymous Reviewers.

I appreciate the work you have done addressing all the reviews and believe the manuscript deserves publication. I only a (very few) minor issues. Please see below an attempt to help.

L.33, "Most climate models [...]" (?)

L.192-196, could you please specify how many GRACE/SMOS observations are assimilated in such one-month window?

We thank the comments from the reviewers. Here we addressed the issues raised by the reviewers below.

Reviewer #1 (Remarks to the Author):

The authors addressed the comments I made in the first article. Specifically they provided a more extensive and detailed explanation of what the model does, how datasets are assimilated, and how the “accessible water storage” is computed. These modifications improved significantly the manuscript.

I appreciate the explanation of methods that is now clearer in the text, but I still think “accessible water storage” is a misleading term because it is solely determined through NDVI anomalies and even though the authors mask regions where the model has low predictive skills, they cannot account for additional “accessible water storage” that does not result in a change in NDVI signal. This is written in the text, however, if somebody goes quickly to the article or worse stop to the results of Figure 2, he/she may interpret the reported value labelled as “accessible water storage” in a quite misleading way.

In response to the reviewer’s concern, we have added the following to Figure 2a caption: “(a) Accessible storage here relates to the soil depth to which vegetation greenness responds most strongly.” We believe this help clarify the definition of ‘accessible storage’.

I might still have a quite different feeling on the practical relevance of the improved prediction skills of NDVI for real problems (now stated at Line 140-150) but I guess, here, we are entering the realms of opinions and the one of the authors is likely as valuable as mine, so I would leave it as it is.

Specific comments

Maybe you can add one sentence in the manuscript to say that the threshold chosen for the correlation does not affect much the results (Fig. 1 in the rebuttal). This is a point in favor of the study.

We thank the reviewer for this comment. We now include the following statement in the method section in L242:

“The specific value of threshold used was based on maximising skilful forecasts. Nevertheless the area of skilful forecasts remains stable with changes in threshold values.”

Line 147. I would suggest to not referring to “vegetation population dynamics”. There is nothing in the article that is directly related to “population dynamics” and while water access surely affects plant competition, it is not this study that can shed light on how different plants access water and compete, what you retrieve it is just an estimated value of water storage over the grid area.

We thank the reviewer for this comment. We have now changed “vegetation population dynamics” to “potential vegetation conditions” in L161.

Line 173. “layers” rather than later.

We thank the reviewer for identifying this typo. It has been corrected.

Reviewer #3 (Remarks to the Author):

Dear Authors,

This is the second time I am reviewing the manuscript. I have now read your responses to my reviews as well as those from two anonymous Reviewers.

I appreciate the work you have done addressing all the reviews and believe the manuscript deserves publication. I only a (very few) minor issues. Please see below an attempt to help.

L.33, "Most climate models [...]" (?)

We understand the reviewers confusion. Here we do mean to use "climate modes" meaning the circulation pattern, and not climate models.

L.192-196, could you please specify how many GRACE/SMOS observations are assimilated in such one-month window?

We thank the reviewer for this suggestion. We specify the observations used in the assimilation window in the manuscript in L212 as:

"A single monthly observation together with all the daily SMOS observations within a one-month window were included in the observation vector."